# Incorporating the Cluster A and V1V2 Targets into a Minimal Structural Unit of the HIV-1 Envelope to Elicit a Cross-Clade Response with Potent Fc-Effector Functions

**DOI:** 10.3390/vaccines9090975

**Published:** 2021-08-31

**Authors:** Rebekah Sherburn, William D. Tolbert, Suneetha Gottumukkala, Andrew P. Hederman, Guillaume Beaudoin-Bussières, Sherry Stanfield-Oakley, Marina Tuyishime, Guido Ferrari, Andrés Finzi, Margaret E. Ackerman, Marzena Pazgier

**Affiliations:** 1Infectious Diseases Division, Department of Medicine of Uniformed Services, University of the Health Sciences, Bethesda, MD 20814, USA; rebekah.sherburn.ctr@usuhs.edu (R.S.); William.tolbert.ctr@usuhs.edu (W.D.T.); Suneetha.Gottumukkala.ctr@usuhs.edu (S.G.); 2Thayer School of Engineering, Dartmouth College, Hanover, NH 03755, USA; Andrew.P.Hederman.TH@dartmouth.edu (A.P.H.); margaret.e.ackerman@dartmouth.edu (M.E.A.); 3Centre de Recherche du CHUM, Département de Microbiologie, Infectiologie et Immunologie, Université de Montréal, Montreal, QC H2X0A9, Canada; guillaume.beaudoin-bussieres@umontreal.ca (G.B.-B.); andres.finzi@umontreal.ca (A.F.); 4Department of Medicine, Duke School of Medicine, Durham, NC 27710, USA; sherry.oakley@duke.edu (S.S.-O.); marina.tuyishime@duke.edu (M.T.); gflmp@duke.edu (G.F.)

**Keywords:** HIV-1, ID2, ID2-V1V2, ADCC, ADCP, Fc-effector functions, vaccines

## Abstract

The generation of a potent vaccine for the prevention and/or control of HIV-1 has been unsuccessful to date, despite decades of research. Existing evidence from both infected individuals and clinical trials support a role for non-neutralizing or weakly neutralizing antibodies with potent Fc-effector functions in the prevention and control of HIV-1 infection. Vaccination strategies that induce such antibodies have proven partially successful in preventing HIV-1 infection. This is largely thought to be due to the polyclonal response that is induced in a vaccine setting, as opposed to the infusion of a single therapeutic antibody, which is capable of diverse Fc-effector functions and targets multiple but highly conserved epitopes. Here, we build on the success of our inner domain antigen, ID2, which incorporates conformational CD4-inducible (CD4i) epitopes of constant region 1 and 2 (C1C2 or Cluster A), in the absence of neutralizing antibody epitopes, into a minimal structural unit of gp120. ID2 has been shown to induce Cluster A-specific antibodies in a BALB/c mouse model with Fc-effector functions against CD4i targets. In order to generate an immunogen that incorporates both epitope targets implicated in the protective Fc-effector functions of antibodies from the only partially successful human vaccine trial, RV144, we incorporated the V1V2 domain into our ID2 antigen generating ID2-V1V2, which we used to immunize in combination with ID2. Immunized BALB/c mice generated both Cluster A- and V1V2-specific antibodies, which synergized to significantly improve the Fc-mediated effector functions compared to mice immunized with ID2 alone. The sera were able to mediate both antibody-dependent cellular cytotoxicity (ADCC) and antibody-dependent cellular phagocytosis (ADCP). We therefore conclude that ID2-V1V2 + ID2 represents a promising vaccine immunogen candidate for the induction of antibodies with optimal Fc-mediated effector functions against HIV-1.

## 1. Introduction

Despite in-depth, detailed knowledge of the Human Immunodeficiency Virus (HIV), no vaccination or passive immunization strategy to date has been successful in preventing HIV-1 infection in humans. HIV-1 is an extremely difficult virus to both prevent and eradicate due to the very restricted exposure of surface proteins for immune recognition and the ability of the virus to integrate into the host genome and remain latent for extended periods of time [1,2]. The only exposed antigen on the surface of the virus and HIV-infected cells is the envelope glycoprotein (Env) [3], and therefore this constitutes the only target available for both broadly neutralizing (bnAbs) and non-neutralizing antibodies (nnAbs) capable of Fc-effector functions to impact the virus. Env is a trimer of transmembrane gp41 and external gp120 polypeptides that is heavily glycosylated and protected from recognition by the immune system. Only after binding to CD4^+^ lymphocytes via the CD4 receptor and co-receptor (CCR5 or CXCR4) do the most majorly conserved epitopes on Env become available for antibody recognition [2]. Due to the high conservation among these epitopes, they make ideal targets for cross-clade antibody induction.

Broadly neutralizing antibodies (bnAbs) against Env have been shown to confer protection against challenge with SHIV and SIV when used as passive transfer agents in a number of Non-Human Primate (NHP) studies [4,5,6,7]. Unfortunately, this success has only been partially replicated in studies in humans; the recent HVTN 704/HPTN 085 trial, which infused a high or low dose of the bnAb VRC01, showed no correlation between treatment groups and the prevention of HIV-1 infection [8]. Vaccination efforts to induce bnAbs have also been unsuccessful, mostly due to the large degree of somatic hypermutation and post translational modification required to elicit an effective bnAb [9,10]. In contrast to the pattern observed for bnAbs, nnAbs capable of Fc-mediated effector responses failed to provide protection from infection in a number of NHP passive transfer studies, which has resulted in this antibody class receiving significantly less attention as a potential treatment strategy for HIV-1 [11,12]. Research into Fc-mediated responses to HIV has mostly focused on the role these antibodies play in individuals in the context of natural infection [13,14,15,16,17]. However, the only partially successful HIV vaccine trial to date (RV144), consisting of a DNA prime and protein boost that showed moderate protection (31.2%) of study participants from HIV-1 infection [18], revealed that nnAb responses elicited by a vaccination was the likely mechanism of protection, which has renewed interest in this antibody class [19,20]. In the trial, antibodies specific for two epitope regions within Env—linear epitopes at the tip of variable loop 2 (V2) and conformational CD4-inducible (CD4i) epitopes within constant region 1 and 2 (C1C2, [13]) of gp120—were associated with vaccine efficacy due to their Fc-effector functions against HIV-1, primarily antibody-dependent cellular cytotoxicity (ADCC) [19,21,22,23]. Follow-up analysis dissecting the antibody response in RV144 subjects revealed antibody-dependent cellular phagocytosis (ADCP) was also associated with HIV-1 protection [24]. ADCP has also been associated with protection from SHIV infection in a macaque challenge study [25].

We recently developed an immunogen referred to as inner domain 2 (ID2), which stably expresses the conformational Cluster A epitope target within a minimal structural unit of gp120 [26]. Immunization studies in BALB/c mice with ID2 demonstrated that ID2 was highly immunogenic and that the Cluster A-specific responses were induced at levels that provided potent Fc-effector activities, including ADCC [27,28,29,30]. Here, we describe the engineering, functional analyses and immunogenicity of ID2-V1V2, an immunogen variant that incorporates both the Cluster A and V1V2 regions of HIV-1 Env to induce Fc-effector function responses to both epitope targets implicated in the protective effect of the RV144 vaccine trial. Immunizations with ID2-V1V2 in combination with ID2 elicited antibodies with greater Fc-mediated responses than ID2 immunization alone. Both ADCC and ADCP were greatly enhanced by the addition of the V1V2 loop sequence. The elicitation of both Cluster A and V1V2 antibody specificities in the absence of other neutralizing Env responses in a single vaccine will permit the testing of the role of Fc-mediated effector functions in the prevention and control of HIV-1 infection.

## 2. Materials and Methods

### 2.1. Immunogen Design and Preparation

ID2-V1V2 consists of residues 44-256 (HXB2 numbering) of clade A/E 93TH057 gp120 connected by a glycine–glycine linker to the C-terminus of gp120 (residues 472-492); in ID2, the V1V2 loop (residues 118-206) is replaced by a glycine–glycine–alanine linker [26]. An extra disulfide was added to stabilize the Cluster A epitope region by mutagenesis of valine 65 and serine 115 to cysteine in both ID2-V1V2 and ID2, sequences are displayed in Appendix A. Initial clones were synthesized with optimized codons for mammalian expression by Blue Heron Biotechnology, Inc., using the native gp120 leader sequence. Mutations were added with the Quickchange PCR mutagenesis kit (Stratagene, San Diego, CA, USA) as per the manufacturer’s protocol and the genes were subcloned into the pACP-tag(m)-2 vector (New England Biolabs, Ipswich, MA, USA) for protein expression.

Plasmids for ID2 or ID2-V1V2 were transfected into HEK 293 cells using EndoFectin Max (GeneCopoeia, Rockville, MD, USA) as per the manufacturer’s protocol. HEK293 cells were grown in FreeStyle 293 expression medium (Gibco, Gaithersburg, MD, USA) at 37 °C in 8% CO_2_ in shaker flasks. Stable cell lines expressing ID2 or ID2-V1V2 were generated from transfected cells using Geneticin (G418 sulfate) (Gibco). Expression cultures of stable cell lines were harvested after 7 days growth under antibiotic selection. Cells were removed by centrifugation and the medium was filtered in preparation for protein purification.

ID constructs were purified using a N5-i5 IgG affinity column [26]. N5-i5 IgG was chemically crosslinked to protein A resin using the Pierce protein A IgG plus orientation kit (Thermo Fisher, Waltham, MA, USA). The N5-i5 IgG affinity column was first equilibrated with PBS pH 7.2. Medium from the ID-expressing cells was passed over the column and the column washed with PBS. Protein was eluted with 0.1 M glycine, pH 3.0, and the eluted fractions were immediately diluted 10:1 with 1 M Tris-HCl, pH 8.5. Elution fractions were combined, concentrated in Amicon ultra centrifugal filters (Millipore, Burlington, MA, USA), and the buffer exchanged to PBS pH 7.2. Concentrated protein was sterile filtered before use in animal studies.

### 2.2. Surface Plasmon Resonance (SPR)

The binding affinity and kinetics of ID2, ID2-V1V2, flsc and gp120 to conformational cluster A and ID2-V1V2 to linear V2-specific mAbs were assessed by surface plasmon resonance (SPR) on a Biacore T-100 (GE Healthcare, Chicago, IL, USA) at 25 °C with an HBS-EP buffer (10 mM HEPES, pH 7.4, 150 mM NaCl, 3 mM EDTA and 0.05% surfactant P-20). Protein A was first immobilized onto the second of two flow cells of a CM5 chip to ~3000 response units (RU) and the first flow cell blocked with standard amine coupling (GE Healthcare). The IgG of interest was captured onto the second flow cell by passage of a 5–10 nM solution of the mAb at 10 µL/min for 30 s. Antibody concentration was varied to give a RU in the range of 100 to 150 RU. ID2-V1V2 was then passed over both flow cells at 30 µL/min for 200 s and allowed to dissociate by passage of buffer over both cells at the same flow rate for 800 s. The flow cells were regenerated by passage of a 0.1 M glycine solution pH 3.0 at 100 µL/min for 60 s and the mAb reloaded onto the second flow cell for the next ID2-V1V2 concentration. Blank subtracted sensorgrams from a two-fold dilution series of ID2-V1V2 starting at 200 nM were used to calculate the kinetic constants using the BIAevaluation software (GE Healthcare, Chicago, IL, USA).

### 2.3. Mice and Immunization

Female and male 6–8-week-old BALB/c mice were purchased from Jackson Laboratories and housed in the Uniformed Services University of the Health Sciences (USUHS) animal facility in accordance with the Association for the Assessment and Accreditation of Laboratory Animal Care International (AAALAC) standard. Groups of 6 mice (3 male, 3 female) were immunized via subcutaneous (SQ) or intraperitoneal (IP) route with ID2 (5 µg) and/or ID2-V1V2 (10 µg). Immunizations were performed at Week 0, 2, 4 and 8 with blood samples collected 2 weeks following each immunization. Terminal exsanguination was performed 2 weeks following the final vaccination. All procedures were approved by the USUHS IACUC committee in advance (Protocol number MED-18-060).

### 2.4. Immunogen-Specific ELISA

Antigen-specific ELISAs were performed to detect the presence of either ID2- or ID2-V1V2-specific antibodies in sera. Firstly, 96-well Nunc Maxisorp plates (Millipore Sigma, Burlington, MA, USA) were coated with 50 ng per well ID2 or an equivalent molar amount of ID2-V1V2 in Tris-buffered saline (TBS) overnight at 4 °C. Plates were then washed with TBS before blocking with TBS + 5% non-fat milk powder and 0.1% Tergitol at room temperature for 2 h. Sera pooled from each immunization group was diluted with blocking buffer in a 10-fold dilution starting at 1:100 and incubated on plates overnight at 4 °C. Plates were then probed with an alkaline phosphatase (AP) secondary antibody (Southern Biotech, Birmingham, AL, USA) for total IgG, IgG1, IgG2a, IgG2b, IgG2c, IgG3, IgA and IgM at a 1:1000 dilution in blocking buffer for 1 h at 37 °C. Plates were washed and developed using a Blue Phos Microwell Phosphatase Substrate System. The reaction was stopped following a 15 min incubation at room temperature using APstop Solution (Both Seracare Life Sciences, Milford, MA, USA). The plates were then read at 620 nm and the optical density recorded. All sera samples were measured in triplicate. GraphPad Prism (Version 7.05, San Diego, CA, USA) was used to display the mean and SEM for all groups and used to calculate the area under the curve (AUC).

### 2.5. Competition ELISA

Antigen-coated plates were prepared as described above. Antibodies to be assessed were biotinylated using the EZ-link Sulpho-NHS biotin kit (Thermo Fisher Scientific, Waltham, MA, USA). The half maximum binding was assessed for each individual antibody and this concentration was mixed 1:1 with diluted sera before plating. Maximum competition was determined by mixing biotinylated antibody with 10 µg/mL of the equivalent unbiotinylated antibody and minimal competition determined by mixing biotinylated antibody with naïve mouse sera. Following overnight incubation at 4 °C, plates were washed and incubated with 1:1000 ExtrAvidin AP (Sigma Aldrich, St. Lewis, MO, USA) for 1 h at room temperature. Plates were developed and read as described above. Competition was determined by normalizing optical density (OD) readings to 100% and 0% competition using GraphPad Prism. All sera were assayed in triplicate and displayed as the mean and SEM for each immunization group.

### 2.6. Rapid Fluorometric Antibody-Dependent Cellular Cytotoxicity

The ADCC activity of the sera immunoglobulins with positive control A32 and negative naïve sera controls were tested with the rapid fluorometric antibody-dependent cellular cytotoxicity (RFADCC) assay [31]. Briefly, GFP-CEM-NKR-CCR5-SNAP cells were coated with a recombinant HIV-1 gp120_Bal_, gp120_93THo57_ or gp120_93TH057_ core, all at 50 μg/mL. Coated target cells were then incubated with diluted sera ranging from 1:50 to 1:51,200, or A32 ranging from 10 to 0.0003 µg/mL. Cells were then mixed with human PBMC effector cells at a ratio of 1:50 target to effector and incubated for 2 h at 37 °C. Following incubation, samples were fixed with paraformaldehyde and collected (at approximately 20,000 events per sample) on a BD LSRII Special Order instrument (BD Biosciences, San Jose, CA, USA) and analyzed using FlowJo software (Tree Star, Ashland, OR, USA). ADCC activity (shown as % cytotoxicity) was defined as the percent of GFP-CEM-NKR-CCR5-SNAP target cells that lost GFP staining but retained the CCR5-SNAP tag staining. The results represent the average of the samples tested in triplicate using a single PBMC donor.

### 2.7. Phagocytosis

ADCP assays were carried out as previously described [32]. Briefly, streptavidin-coated 1 µm fluorescent microspheres were coated with biotinylated FLSC overnight at 4 °C. Following washing, the beads were incubated with diluted sera for 2 h at 37 °C, and the sera were analyzed in triplicate. For Monocyte ADCP, THP-1 cells were utilized as effectors, and for ADNP, neutrophils isolated from two healthy individuals were used. Cells were added to the bead/antibody mixture and incubated overnight to allow phagocytosis. Samples were then fixed and analyzed via flow cytometry in triplicate to define the fraction and fluorescent intensity of cells that phagocytosed one or more bead.

### 2.8. Complement Deposition

Antibody-dependent complement deposition assays were carried out as previously described [33]. Briefly, fluorescent neutravidin beads were conjugated to biotinylated FLSC and incubated with sera diluted 1:10, 1:100 and 1:250. Immune complexes were then incubated with guinea pig complement (Cedarlane, Burlington, ON, Canada) for 20 min at 37 °C, followed by washing. Fluorescein-conjugated goat IgG directed at C3 (MpBio, Irving, CA, USA) was used to detect complement deposition. Beads were acquired using a BD LSR II plate reader with results representative of two separate runs.

### 2.9. Statistical Analysis

Statistical differences between the ID2 and ID2-V1V2 samples were determined by either by 2-way ANOVA or 2-way ANOVA with Sidak multiple comparison correction. All statistical analyses were carried out using GraphPad Prism software (San Diego, CA, USA), with significance indicated as * *p*  <  0.05, ** *p*  <  0.01, *** *p*  <  0.001 or **** *p*  <  0.0001.

## 3. Results

### 3.1. V1V2 Is Presented in the Native Conformation When Co-Expressed with ID2

RV144 and follow-up NHP studies indicated a synergy between Cluster A and V2 tip-specific Abs in driving the most effective responses in vivo [23]. mAbs specific for the Cluster A and V2 regions isolated from RV144 subjects synergized to mediate potent ADCC against cells infected with both tier 1 and 2 isolates defective for Nef expression. In addition, virus sieve analysis detected a vaccine-induced site of immune pressure at position 169 in the V2 loop (169K) [34]. The ADCC activity of V2 specific mAbs were dependent on this position in breakthrough Envs [21,34]. The epitopes for RV144 V2 Abs map to linear sequences at the V2 loop tip, around residues 168–183 of gp120 [21,35], partially overlapping the V2i integrin binding site and quaternary V2q epitope recognized by bnAbs PG9 and PG16. This V2 region shows high sequence variability and structural studies confirm that it may adopt several divergent conformations [21].

Our successful design of stable ID2 [26] prompted us to develop a variant incorporating both the Cluster A and V1V2 regions, therefore generating an immunogen that would selectively present these two important vaccine targets. The ID2 molecule, stabilized to maintain the CD4-triggered ID conformation, constitutes an optimal Env domain suitable for V1V2 loop incorporation and presentation in the same conformation as the full-length gp120. Moreover, the initial RV144 V2 IgG correlate was identified using a scaffolded V1V2 construct (gp70-V1V2 fusion protein) and no correlation was observed with linear and cyclized V2 peptides [20,21]. This points toward the possibility that although linear, the ADCC V2 epitopes may be more accessible for Ab recognition when the V1V2 loop is scaffolded on a large protein platform. We therefore generated ID2-V1V2 by engineering the V1V2 loop (clade A/E 93TH057, residue 118-206) at the ID2–GGA linker (Figure 1A); the DNA and amino acid sequence are shown in Appendix A. We developed a stable cell expression line producing about 1 mg of ID2-V1V2 per liter of culture. The ID2-V1V2 construct contains 14 cysteine residues; 8 within the ID2 design and 6 within the V1V2 region. A single peak for both immunogens was detected by gel filtration (Figure 1B) and a single band was detected by SDS-Page (Figure 1C) migrating as glycosylated protein with a molecular weight of ~60 kD, confirming that all cysteines were involved in disulfide bond formation and purified ID2-V1V2 is folded, fully oxidized and consisted of a homogeneous species in solution and was not a mixture of different disulfide-bonded isomers.

To determine if the desired epitopes within the Cluster A and V2 regions were fully preserved in ID2-V1V2, we analyzed the antigenic properties of the immunogen by binding kinetics to a panel of Cluster A mAbs and V2-specific antibodies using SPR (Figure 2A,B) and ELISA (Figure 2C). In the V2 Ab panel we used V2i/p-specific (697, CH58 and HG120) and V2q-specific (PG9 and PG16) mAbs. ID2-V1V2 displayed high binding affinities for Cluster A mAbs with K_D_ values in the range of 0.7–9.3 nM (similar to affinities FLSC [36,37]); however, the antibodies bound with higher affinity to ID2. Binding affinities of the V2-specific antibodies CH58 and HG120 were comparable to those described before for the RV144 immunogen, AE.A244Δ11 gp120 [23]. PG9/16 Abs did not bind to ID2-V1V2 by SPR. As shown in Figure 2C, of the V2 Abs tested by ELISA, all but PGT145 bound to ID2-V1V2, with the highest binding observed for Vp Abs: CH58, CH59, Cap228-19K/F [38] and V2i-specific Ab 697, and lower but detectible binding to the V2q-specific Ab PG16. All together, these studies indicate that ID2-V1V2 adopts the CD4-bound conformation in solution, as evidenced by increased affinity for Cluster A mAbs compared to gp120. Importantly, ID2-V1V2 also presents the V1V2 epitopes recognized by RV144 V2-specific Abs.

### 3.2. ID2, ID2-V1V2 Mix Elicits More Antigen-Specific Antibodies than ID2 Alone

To investigate the immunogenicity of ID2-V1V2, we first immunized six BALB/c mice per group with either ID2 (5 µg) or ID2-V1V2 (10 µg) supplemented with GLA-SE as an adjuvant with three doses two weeks apart, followed by a final boost 4 weeks after the third dose with a protocol similar to that used previously for ID2 immunizations alone [39]. Modifications to the protocol included a change of the immunization route from Intraperitoneal immunization (IP) to Subcutaneous (SQ). Unfortunately, the SQ vaccine route resulted in poor induction of Cluster A responses with ID2 immunization as compared to past IP immunizations (Appendix A). Analysis of the ID2-V1V2 sera revealed that responses to Cluster A targets also were consistently lower when ID-V1V2 was used alone in immunization, leading to a lack of A32 competition in ELISA and a poor functional response, including weak RFADCC and ADCC (Appendix A). However, we were able to detect V1V2 responses, as indicated by the competition ELISA (Appendix A), pointing toward the possibility that the lower antibody affinities seen by SPR to ID2-V1V2 translated into poor accessibility of the Cluster A region (Figure 2A) when injected into mice. Poor Cluster A responses were likely due to the large size of the V1V2 loop either masking the epitope region due to heavy glycosylation or the added flexibility added around the stabilizing disulfide bond.

To mitigate the problem of potential destabilization of the conformational Cluster A epitope targets within the ID2-V1V2 immunogen, we modified our immunization composition to include a mix of the ID2 and ID2-V1V2 immunogens. Groups of six Balb/c mice were immunized IP with ID2 alone (5 µg) or ID2 (5 µg) mixed with ID2-V1V2 (10 µg) with GLE-SE as the adjuvant, according to the immunization scheme shown in Figure 3A. This protocol allowed immunizations with ID2 (17 KDa) at doses identified previously as optimal for Cluster A response induction alongside with a similar molar ratio of V1V2 (34 KDa) scaffolded on the ID2 platform.

Sera from immunized mice were then assayed for antigen-specific total IgG over the course of the vaccination protocol and for multiple antigen-specific isotypes in terminal sera (Figure 3B,C). This was achieved by assessing binding against equal molar amounts of ID2 (ID2 alone immunization) or ID2-V1V2 (ID2 + ID2-V1V2 immunization). Immunization with ID2 + ID2-V1V2 resulted in significantly increased titers of IgG early in the vaccination protocol as well as at the termination (Figure 3B). Interestingly, the distribution of antigen-specific isotypes elicited by both immunogens was different, with ID2 + ID2-V1V2 immunization leading to higher titers of IgG1, IgG2a, IgG2b and IgA, while ID2 alone resulted in higher IgG2c and slightly elevated IgG3 and IgM (Figure 3C).

### 3.3. ID2-V1V2 Immunization Generates Sera Capable of Blocking the Binding of V1V2 Antibodies

In order to determine if both Cluster A and V1V2 antibodies had been elicited in response to the ID2 + ID2-V1V2 immunogens, we tested sera for competition against prototype antibodies to both ID2 and V1V2. Sera from both immunizations was tested against binding to ID2 in competition with both A32 (Figure 4A) and N5-i5 (Figure 4B). Both sera resulted in equal competition of A32 and N5-i5, demonstrating the successful elicitation of Cluster A antibodies by ID2. The equal competition in each group again shows that it is likely that the A32 and N5-i5 epitopes in ID2-V1V2 are destabilized by the addition of the V1V2 loop, as no additional Cluster A competition is seen from the inclusion of 10 µg ID2-V1V2. The elicitation of V1V2-specific antibodies was confirmed by the successful binding of sera to ID2-V1V2 when in competition with CH59, CAP228F and CAP228K (Figure 4C). These antibodies are all V2 tip specific; CH59 was isolated from an RV144 vaccine [21], while CAP228F and CAP228K were isolated from an HIV-1-infected donor [38].

### 3.4. V1V2 Enhances ADCC against Cross-Clade gp120-Coated Target Cells

Next, we aimed to determine if the addition of V1V2-specific antibodies to the Cluster A-specific responses induced by ID2 alone would boost the Fc-effector function of sera. First we assessed ADCC using the rapid fluorometric ADCC (RFADCC) protocol, described in [31], where sera were tested for the ability to lyse CD4-positive cells coated with gp120. We used matched clade AE (gp120_93TH023_ core and gp120_93TH023_) and immunogen mismatched, clade B (gp120_BAL_). As shown in Figure 5A, ADCC against core-bound CEM-Nkr cells was very similar between both immunization groups, with peak lysis reaching 92.7% for ID2-V1V2 sera and 86.7% for ID2 sera when compared to the A32-positive control. In contrast, a slight increase in maximal lysis was observed with the ID2-V1V2 sera for cells coated with full-length monomeric gp120_93TH057_, which peaked at a sera dilution of 1:200 compared to ID2 alone, with the highest response at a 1:50 sera dilution (Figure 5B). Interestingly, the level of ADCC induced by ID2-V1V2 sera was not affected by a mismatched clade of gp120. Bal gp120-coated CEM-Nkr cells were lysed equally when incubated with sera from ID2-V1V2-immunized mice, with a comparable Max lysis (63.5% for a clade match vs. 57.5% for clade mismatch) and AUC (Both 1.24 *×* 10^5^). The same pattern was not observed with ID2 sera as the clade matched to ADCC (max. lysis 52%, AUC 6.3 *×* 10^4^) was more potent than the mismatched clade (max. lysis 38%, AUC 4.1 *×* 10^4^) (Figure 5C).

### 3.5. ID2-V1V2 Immunization Elicits FLSC-Specific Antibodies Capable of Phagocytosis

Finally, we wanted to test the potential of immunized sera to induce antibody-mediated phagocytosis (ADCP) of beads coated with a recombinant Env antigen. In the assay we used a full-length single chain (FLSC), a covalent dimer of full-length gp120, of clade B, and domain 1 and 2 of human CD4 [40]. FLSC expresses the Cluster A and V1V2 epitopes in the same conformation as CD4-trigered gp120. Analysis of sera binding to FLSC showed ID2-V1V2 elicited more FLSC-specific antibodies than ID2 alone (OD 1.27 vs. 0.69 at a 1:100 dilution, respectively) (Figure 6A). This translated to significantly higher levels of ADCP of the FLSC-coated bead for all tested dilutions of ID2-V1V2 sera by THP-1 cells, derived from monocytes, but did not have an impact on ADNP by neutrophils (Figure 6B,C). As seen in ADCC, the monocyte phagocytosis assay displays a prozone-like effect where a higher concentration of sera resulted in reduced effector cell function. This is very clearly seen in a number of published HIV ADCC studies, dating back almost 30 years [41], and more recently for ADCP studies [42]. The increased FLSC-specific antibodies also did not translate to higher ADCD, with complement deposition higher in the sera from the ID2-only immunized mice (Figure 6D). Both ADNP and ADCD have been shown to be dependent on human IgA; however, how this translates to mouse IgA used in this study is unknown [43]. Therefore, despite inducing antibodies to small, defined epitopes of gp120, sera from ID2 + ID2-V1V2 is capable of mediating ADCP against full-length antigens presenting Env in the CD4-bound conformation.

## 4. Discussion

Existing evidence supports the notion that targeting a single Env epitope by vaccine-induced or passively transferred antibodies has little or no chance for success in the prevention or control of HIV-1 infection. The recent bnAb human passive immunization trials HVTN 704/HPTN 085 clearly showed this as escape variants emerged quickly to overcome immune pressure at a single Env site [8]. It is therefore also likely that any therapeutics using passive transfer of non- or weakly-neutralizing antibodies, which rely on Fc-effector activities for their function, will fail if only a single epitope is targeted. In addition to the requirement for multiple epitope targets to prevent virial escape, Fc-effector mechanisms benefit from the synergy between multiple epitope specificities for enhanced effector cell recruitment and Fcγ receptor triggering [44]. The classic example of such synergy is the V2 tip and Cluster A-specific antibodies induced in the RV144 vaccine trial that synergized to mediate the most effective infectious tier 1 and 2 virus capture and ADCC [18,22,23,35,45]. Interestingly, ADCC and anti-V1/V2 Ab induction was also shown to correlate with reduced infection risk in several NHPs studies replicating RV144 [20,46]. The V2-directed antibodies isolated from RV144 participants map to linear sequences at the V2 loop tip, around residues 168–183 of gp120 [21,35], partially overlapping the V2i integrin binding site and quaternary V2q epitope recognized by bnAbs PG9 and PG16. This V2 region shows high sequence variability and structural studies confirm that it may adopt several divergent conformations [21]. Abs recognizing this region show weak neutralizing activity and are capable of potent Fc-effector functions, including ADCC and ADCP, in a macaque model [47]. Similarly, Cluster A epitopes are known targets for Abs capable of potent ADCC, ADCP and antibody-dependent cellular trogocytosis (ADCT); however, in the absence of direct neutralization activity. These targets are also referred to as Cluster A epitopes [13,48]; the A32 and C11 antibodies isolated from HIV-infected individuals are prototype mAbs of this specificity [37,49,50]. In context of the pre-fusion, closed Env trimer decorating free HIV-1 particles, the Cluster A targets are occluded and therefore not accessible for Ab recognition. These epitopes are strictly CD4-dependent and become exposed only upon interaction of the Env trimer with host cell CD4. They can be detected during the entry process when virions attach to uninfected cells [48,51,52,53,54,55], on the surface of HIV-infected cells retaining detectable CD4 (CD4+ cells) [15,56,57,58] and in the process of cell-to-cell transmission [59,60].

Here, we describe the development of an immunogen selectively expressing Cluster A and V1V2 epitope targets within a minimal structural unit of HIV-1 Env. We aimed to preserve a native conformation of the V1V2 region; therefore, we modified our existing ID2 immunogen to incorporate the entire V1V2 loop region. Our ID2 immunogen consists of the gp120 inner domain only stabilized in CD4-bound conformation by extra disulfide bonds [26] and could be modified easily to incorporate the V1V2 loop (clade A/E 93TH057, residue 119–210) at the V1V2 stem (–GGA linker, Figure 1). Initial biochemical characterization confirmed the integrity of ID2-V1V2; however, SPR data revealed weaker binding of Cluster A antibodies A32 and N5-i5. ELISA binding to the prototype antibodies of the V2 region confirmed its proper antigenicity.

Immunization with ID2 + ID2-V1V2 resulted in efficient induction of both the Cluster A- and V2-specfic response, as indicated by sera competition ELISA with prototype antibodies of both epitope regions. Analysis of Fc-effector functions with immunized sera showed slightly enhanced ADCC against clade-matched gp120-coated targets and significantly enhanced ADCC against cross-clade targets. In addition to enhanced ADCC, the ADCP against FLSC-coated beads was also enhanced when both Cluster A and V1V2 antibodies were elicited. Chung et al. (2015) highlighted the potential role for ADCP in the protection from HIV-1 using in-depth systems serology to dissect vaccine-induced protection. The group determined that ADCP was enhanced in RV144 vaccines with high V1V2-specific Abs and concluded that ADCP may play a critical role in protection from mucosal transmission of HIV [24]. FLSC consists of Bal gp120 fused to the D1D2 domain of CD4 [40], and therefore effectively displays both Cluster A and V1V2 antigens in the CD4-bound conformation. Clinical trials for immunization with FLSC have completed phase I and II, with promising results for safety and efficacy. Both ADCC and ADCP assays used in this study rely on human monocytes as effectors for murine antibodies. Mouse IgG2 in particular is known to bind with high affinity to human monocytes [61] and induction of this isotype was greatly enhanced with the addition of V1V2 epitopes in the immunization. The species mismatch in effector cell assays is likely to underestimate the effectiveness of the vaccine in a real-world setting.

## 5. Conclusions

We were able to improve upon our successful minimal immunogen ID2, which displays epitopes for the induction of nnAbs in the absence of neutralizing epitopes. By incorporating the V1V2 region we successfully recapitulate the findings of RV144 and multiple NHP models showing that antibodies to multiple epitope regions (in particular, Custer A and V2) likely synergize to enhance Fc-mediated effector functions. ID2-V1V2 + ID2 therefore constitutes a promising vaccine candidate to investigate the effect of nnAbs in the absence of nAbs for the prevention and control of HIV-1 infection.

## Figures and Tables

**Figure 1 vaccines-09-00975-f001:**
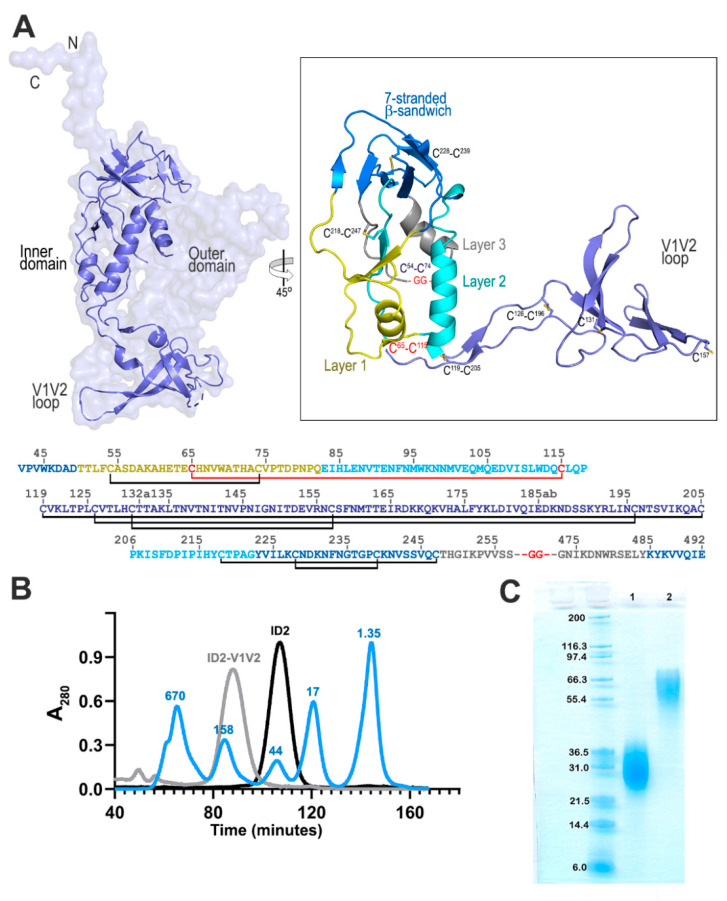
ID2-V1V2 immunogen design and its putative structure. (**A**) Ribbon diagram of the gp120 inner domain and V1V2 loop as is present in the SOSIP trimer with gp120 shown as a surface (left). Putative structure of ID2-V1V2 generated from the structure of ID2 (PDB ID 4YC2) and the V1V2 loop from a CD4 and 17b bound gp120 model (PDB ID 3J70) (right). The “Layered” architecture of the gp120 inner domain is shown with the 7-stranded β-sandwich colored blue, layer 1 yellow, layer 2 cyan, layer 3 gray and the V1V2 loop blue-violet. The ID2-V1V2 sequence is shown below with predicted disulfide bonds present in SOSIP gp120 (black) and ID2 (red). In ID2-V1V2, the gp120 outer domain is replaced by a Gly-Gly linker shown in red. (**B**) Overlay of the gel filtration chromatographs of ID2-V1V2, ID2, and the protein standards, labeled in kDa. (**C**) SDS-PAGE of ID2 (lane 1) and ID2-V1V2 (lane 2) with the protein standards, labeled in kDa.

**Figure 2 vaccines-09-00975-f002:**
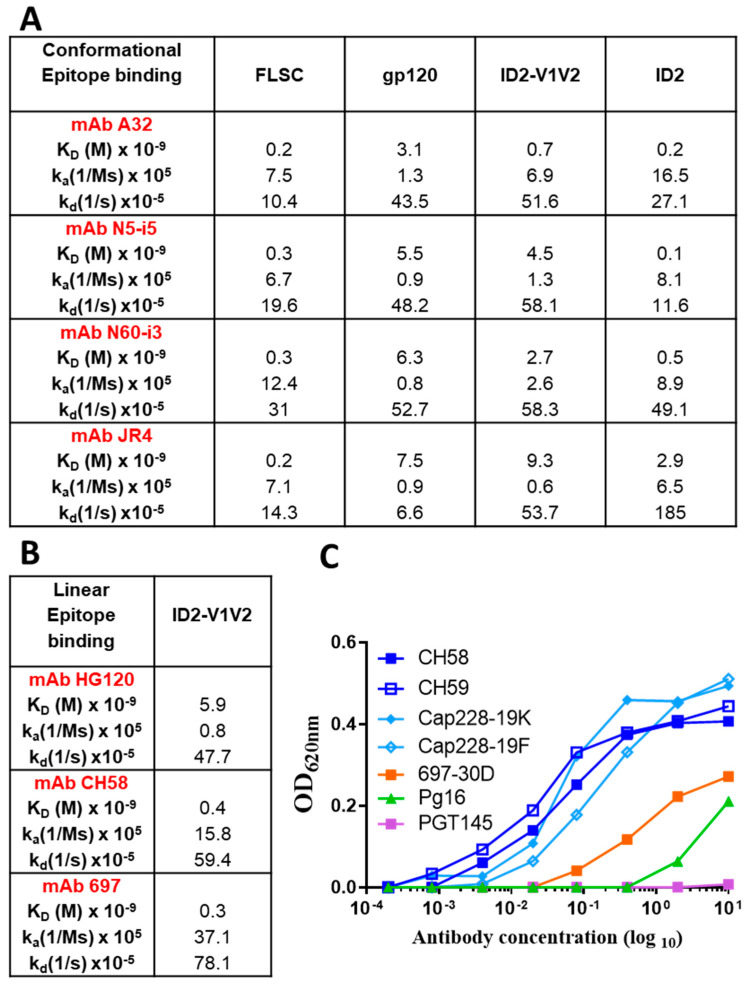
Characterization of ID2-V1V2. (**A**) Binding kinetics of conformational Cluster A Abs to the FLSC, gp120 and ID2-V1V2 by SPR. (**B**) Binding kinetics of linear V2 Abs to ID2-V1V2 by SPR. Abs were captured onto A Protein A CM5 sensor chip and antigens passed over at 0–200 nM concentrations. The binding kinetics (association rates (ka), dissociation rates (kd) and affinity constants (Kd)) were calculated with the BIAevaluation software. (**C**) ELISA showing binding of multiple V2-specific antibodies to 50 ng per well adsorbed ID2-V1V2 over a range of Ab concentrations from 10 µg to 0.00012 µg.

**Figure 3 vaccines-09-00975-f003:**
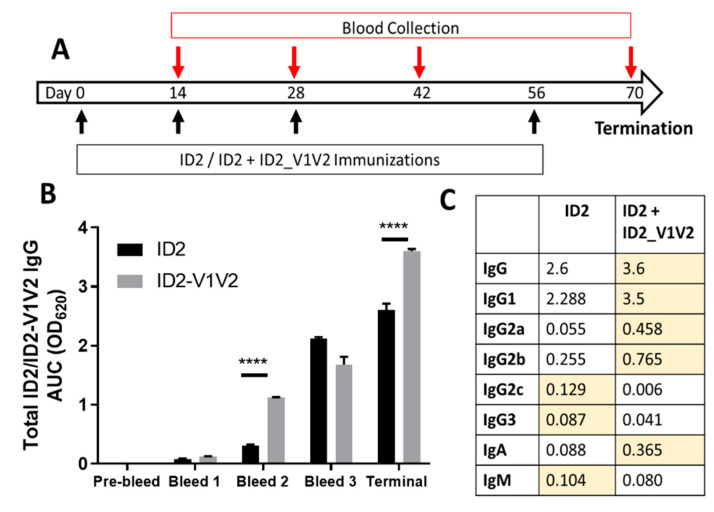
ID2_V1V2 immunogenicity studies in BALB/c mice. (**A**) Vaccination strategy. Mice were vaccinated with either ID2 (5 µg/mL) or ID2 + ID2-V1V2 (5 µg + 10 µg) at Day 0, 14, 28 and 56 with blood collected at Day 14, 28, 42 and 70. (**B**) Total antigen-specific IgG over the time course of vaccination for ID2 alone and ID2 + ID2-V1V2-immunized mice, displaying the area under the curve with statistical analysis via two-way ANOVA to detect differences between the ID2 and ID2-V1V2 samples; **** *p*< 0.0001. Sera were assayed in triplicate with the mean ± SEM displayed (**C**) AUC of the antigen-specific sera in the terminal bleed for multiple isotypes. Higher samples are represented in yellow.

**Figure 4 vaccines-09-00975-f004:**
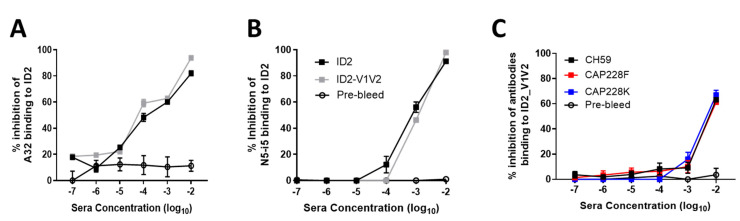
Competition ELISA. Sera from mice immunized with ID2 (grey) or ID2 + ID2-V1V2 were assessed for competition, with (**A**) A32 and (**B**) N5-i5 against plate-bound ID2. (**C**) Sera from ID2-V1V2-immunized mice were assessed for competition, with CH59, CAP-228F and CAP-228K against plate-bound ID2-V1V2.

**Figure 5 vaccines-09-00975-f005:**
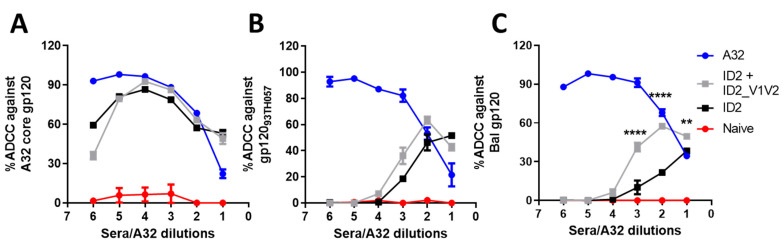
Sera RFADCC. A32 and sera were assessed for ADCC activity using the RFADCC assay against target cells coated with (**A**) gp120_93TH057_ core, (**B**) gp120_93TH057_ and (**C**) gp120_Bal._ Sera was diluted 1:50 and A32 to 10 µg/mL for dilution 1; both were then serially diluted 1:4 to form a dilution curve. Sera and A32 were analyzed in triplicate, with one PBMC donor, and displayed as the mean ± SEM. Statistical analysis was via two-way ANOVA with Sidak multiple comparison correction to detect differences between the ID2 and ID2-V1V2 samples; ** *p*  <  0.01, **** *p * <  0.0001.

**Figure 6 vaccines-09-00975-f006:**
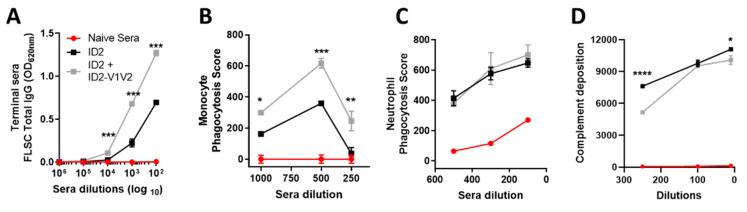
Fc-effector functions against a full-length single chain. (**A**) Full-length single chain (FLSC)-specific IgG induced in the sera of ID2 and ID2+ID2-V1V2 immunized mice. (**B**) Antibody-dependent cellular phagocytosis (ADCP) by THP-1 effectors and (**C**) antibody-dependent neutrophil phagocytosis (ADNP) of the FLSC-coated beads incubated with immunized sera, using two separate neutrophil donors. (**D**) Antibody-dependent complement deposition (ADCD). All analysis was carried out in triplicate, displaying the mean ± SEM. Statistical analysis is via two-way ANOVA with Sidak multiple comparison correction to detect differences between ID2 and ID2-V1V2 samples; * *p*  <  0.05, ** *p*  <  0.01, *** *p*  <  0.001, **** *p*  <  0.0001.

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
