# Peer review of "Incorporating the Cluster A and V1V2 Targets into a Minimal Structural Unit of the HIV-1 Envelope to Elicit a Cross-Clade Response with Potent Fc-Effector Functions"

_vaccines, 2021, doi:10.3390/vaccines9090975_

Round 1
Reviewer 1 Report
To the authors:
The review by Sherburn et al. demonstrated the immunogenicity of a HIV-1 (ID2-V1V2) vaccine, generating non-neutralizing antibodies. As a result, antibodies from ID2-V1V2 immunized BALB/c mice had an effective Fc-mediated effector function in immune cells by mediating antibody dependent cytotoxicity and antibody dependent phagocytosis. The following concerns should be addressed.
Major concern:
- Figure 6B: The reviewer suggests that instead of using THP-1 for induced antibody mediated phagocytosis, the application of primary human monocytes from peripheral blood should be used to verify the findings obtained from THP-1.
Minor Concerns:
- Abstract: Both ADCC and ADCP should be defined.
- Figure legend 6: “chin” should be chain.
- Figure 6 is not sufficiently described or discussed. For instance, why does a high antibody concentration in the serum provoke a lower phagocytosis score in THP-1, in contract to a lower concentration?
Author Response
Reply to reviewer 1
The review by Sherburn et al. demonstrated the immunogenicity of a HIV-1 (ID2-V1V2) vaccine, generating non-neutralizing antibodies. As a result, antibodies from ID2-V1V2 immunized BALB/c mice had an effective Fc-mediated effector function in immune cells by mediating antibody dependent cytotoxicity and antibody dependent phagocytosis. The following concerns should be addressed.
We thank the reviewer for the favorable summary of the paper. Please find below a point by point response to any concerns raised. All changes to the text have been added as ‘track changes’.
- Figure 6B: The reviewer suggests that instead of using THP-1 for induced antibody mediated phagocytosis, the application of primary human monocytes from peripheral blood should be used to verify the findings obtained from THP-1.
We thank the reviewer for this valuable suggestion to improve the paper. This assay utilizing THP-1 cells was originally developed by our collaborator and co-author Margaret Ackerman [1]. One follow-up paper from this lab investigated the phagocytic ability of both primary monocytes and THP-1 cells utilizing antibodies isolated from HIV-1+ subjects. The paper described the outcome, and below is the graph showing good correlation between both cell types [2]. There are also numerous other published papers (>10) which utilize both cell types in a number of disease settings and find comparable results. We therefore hope that the reviewer is satisfied with the utilization of this cell line for the purposes of this paper.
- Abstract: Both ADCC and ADCP should be defined.
Thank you for noticing this, the acronyms have been expanded
- Figure legend 6: “chin” should be chain.
Thank you for noticing this, it has been corrected
- Figure 6 is not sufficiently described or discussed. For instance, why does a high antibody concentration in the serum provoke a lower phagocytosis score in THP-1, in contract to a lower concentration?
We thank the reviewer for picking up on this. In many of the Fc-effector assays utilized, a prozone-like effect (also known as the Hook-effect) is seen where higher concentrations of antibodies or antigen lead to reduced effector cell functions. It is hypothesized that the excess antigen/antibody impairs the ability of antibodies to form effective immune complexes, leading to artificially low results in high concentrations. This is very clearly seen in a number of published HIV ADCC studies, dating back almost 30 years [3], and more recently for ADCP studies [4]. Text has been added using ‘track changes’ to the results and discussion to expand on the findings of figure 6, and to describe the prozone effect seen in our fc-effector assays throughout the paper.
References
- Ackerman, M.E., et al., A robust, high-throughput assay to determine the phagocytic activity of clinical antibody samples. J Immunol Methods, 2011. 366(1-2): p. 8-19.
- Ackerman, M.E., et al., Enhanced phagocytic activity of HIV-specific antibodies correlates with natural production of immunoglobulins with skewed affinity for FcgammaR2a and FcgammaR2b. J Virol, 2013. 87(10): p. 5468-76.
- Jenkins, M., J. Mills, and S. Kohl, Natural killer cytotoxicity and antibody-dependent cellular cytotoxicity of human immunodeficiency virus-infected cells by leukocytes from human neonates and adults. Pediatr Res, 1993. 33(5): p. 469-74.
- Duchemin, M., et al., Antibody-Dependent Cellular Phagocytosis of HIV-1-Infected Cells Is Efficiently Triggered by IgA Targeting HIV-1 Envelope Subunit gp41. Front Immunol, 2020. 11: p. 1141.

Reviewer 2 Report
In this manuscript, Sherburn and colleagues generate and characterize a synthetic immunogen, termed ID2-V1V2, which was designed by fusing Cluster A and V1V2 regions of the gp120 protein of HIV. The authors engineered ID2-V1V2 based on their previously described immunogen, ID2. The authors confirmed the correct production of the ID2-V1V2 peptide using gel filtration chromatography and assess its antigenic properties using a panel of monoclonal antibodies known to be specific for Cluster A or V1V2 regions. The authors measured immunogenicity of ID2-V1V2 construct in vivo by injecting the protein to BALB/c mice. The authors used sera from injected mice to measure the generation of different subclasses of antigen-specific Igs. An in vitro model was also used to test ability of the sera to induce ADCC and ADCP.
The manuscript is of interest and the findings may be relevant for clinical applications. However, there are minor technical concerns that needs to be addressed before publication by the authors.
Comments:
- Although the manuscript is in general readable and well written, there are some typos and some figure legend missing. Some examples: lines 40 (do?), 154, 179/180, 252, 323, 367, 369, A-B missing in figure 2, figure 4A missing color legend. The abstract needs to be edited to make the findings more readable to a broader scientific audience.
- The materials and methods section should be revised. For example, a paragraph describing the SPR method is missing. Furthermore, there are two distinct paragraphs describing different ADCC methods (line 155 and line 189), whereas one of these seems not to be employed in the results section of the manuscript.
- In figure 1A, a full-length representation of gp120 protein with highlighted the regions of interest that were used to design ID2-V1V2 may be of help to better understand how the immunogen was conceived. In Figure 1B, axis labels are missing.
- Figure 2A could be improved. Showing the affinities and binding kinetics of all the mAbs for all the proteins investigated would make the table easier to read and more informative. Moreover, in the result section authors refer to mAbs PG9 and PG16, but these are not showed in the table.
- In figure 3, a negative control is missing. Authors could include sera form adjuvant-injected or naïve mice.
- In figure 4, as in Figure 3, a negative control is missing. Authors could include sera form adjuvant-injected or naïve mice.
- Figure 5, x axis label is not clear. The authors should incorporate a label that enables identification of both the mAb A32 and sera.
- Type and number of replicates are not indicated in the figure legends.
Author Response
Response to reviewer 2
In this manuscript, Sherburn and colleagues generate and characterize a synthetic immunogen, termed ID2-V1V2, which was designed by fusing Cluster A and V1V2 regions of the gp120 protein of HIV. The authors engineered ID2-V1V2 based on their previously described immunogen, ID2. The authors confirmed the correct production of the ID2-V1V2 peptide using gel filtration chromatography and assess its antigenic properties using a panel of monoclonal antibodies known to be specific for Cluster A or V1V2 regions. The authors measured immunogenicity of ID2-V1V2 construct in vivo by injecting the protein to BALB/c mice. The authors used sera from injected mice to measure the generation of different subclasses of antigen-specific Igs. An in vitro model was also used to test ability of the sera to induce ADCC and ADCP.
We thank the reviewer for the favorable summary of the paper. Please find below a point by point response to any concerns raised. All changes to the text have been made as ‘track changes’
- Although the manuscript is in general readable and well written, there are some typos and some figure legend missing. Some examples: lines 40 (do?), 154, 179/180, 252, 323, 367, 369, A-B missing in figure 2, figure 4A missing color legend. The abstract needs to be edited to make the findings more readable to a broader scientific audience.
We thank the reviewer for the thorough examination of the paper and appreciate the highlighted typos and missing information. All typos have been corrected using ‘track changes’ to make the alterations easy to identify. I couldn’t identify the typo on line 40.
For clarity, Figure 2 A has been altered to include two separate tables, one describing structural epitopes and one linear. This figure now includes the labeling for A, B and C.
As with other figures, those with identical colouring are only described once per figure, so figure 4A has not been altered due to the legend being the same as in figure 4B.
A few sentences were added to the abstract to outline the broader applicability of the paper, making it more relatable to a larger audience.
- The materials and methods section should be revised. For example, a paragraph describing the SPR method is missing. Furthermore, there are two distinct paragraphs describing different ADCC methods (line 155 and line 189), whereas one of these seems not to be employed in the results section of the manuscript.
We thank the reviewer for catching this omission. SPR methods have now been added.
For the ADCC methods, both methods are used for the supplementary figure. I have therefore removed some of the text from the main text and included it in the zip file for the supplementary.
- In figure 1A, a full-length representation of gp120 protein with highlighted the regions of interest that were used to design ID2-V1V2 may be of help to better understand how the immunogen was conceived. In Figure 1B, axis labels are missing.
We thank the reviewer for the suggestion of showing ID2 / ID2-V1V2 in the context of gp120 and agree that this figure would benefit the readers in understanding the origin of the immunogen. Figure 1 has therefore been updated to include this schematic.
Figure 1B now contains the axis labels and molecular weights for the standards.
The figure legend has been updated
On the advice of another reviewer we have also added a supplementary figure showing the DNA and aa sequence of ID2-V1V2.
- Figure 2A could be improved. Showing the affinities and binding kinetics of all the mAbs for all the proteins investigated would make the table easier to read and more informative. Moreover, in the result section authors refer to mAbs PG9 and PG16, but these are not showed in the table.
We thank the reviewer for this suggestion, and again we agree that this change will aid the readability and understanding of the manuscript. We have therefore made it clear that ID2, ID2-V1V2, FLSC and gp120 were analysed using conformational antibodies, in order to check the correct folding of the cluster A region. Whereas antibodies to V2 were only tested against ID2-V1V2 because these are linear antibodies and do not depend upon correct folding. To make this more clear, we have split the table into two, one showing conformational epitopes and one linear. We have also removed the fold change, as we felt it confused the data.
PG9 and PG16 were tested for binding via SPR, but did not bind, as stated in the text. Weak binding of PG16 was seen by ELISA in figure 2C.
- In figure 3, a negative control is missing. Authors could include sera form adjuvant-injected or naïve mice.
We thank the reviewer for this observation. Although there was no antigen specific signal detected in naïve mice, we included this in the graph to make this clear to the reader by including a ‘pre-bleed’ column.
- In figure 4, as in Figure 3, a negative control is missing. Authors could include sera form adjuvant-injected or naïve mice.
Thank you for noticing this, the pre-bleed data has now been included in all competition plots.
- Figure 5, x axis label is not clear. The authors should incorporate a label that enables identification of both the mAb A32 and sera.
We thank the reviewer for pointing this out. The axis has been changed to read ‘sera/A32 dilutions’ and the dilution of both sera and A32 explained in the figure legend.

Reviewer 3 Report
Summary
Sherburn et al. design a minimal HIV envelope construct to induce potent Fc dependent responses s. The authors previously presented data on a minimal structural unit comprising the inner domain antigen (ID2), a region that includes the CD4 induced epitope (i.e. upon CD4 binding the epitope becomes available, CD4i) from the constant region 1 & 2 (C1, C2) which is also referred to as region A. Some data suggests that in so far only partially successful vaccine trial RV144 with the V1V2 epitope, the antibody dependent cellular cytotoxicity (ADCC) by non-neutralizing antibodies (nnAbs) plays a role in reducing HIV infection. Likewise, the ID2 epitope has been shown by the authors to elicit non-neutralizing antibodies that induce strong Fc dependent responses. This suggests that both strategies result non-neutralizing antibodies that trigger ADCC dependent responses. Here, they reasoned that the two epitopes that were partially successful, ought to be combined in a structure that is based on their ID2. To this end, they engineered the V1V2 on top of their ID2 structural unit. The ID2-V1V2 showed poor responses to ID2, suggesting that the engineered V1V2 loop might shield the ID2 scaffold. However, they find that immunization with ID2-V1V2 together with ID2 induced a strong Fc-dependent response. The ADCC response was stronger than with their previous ID2 alone. Moreover, antibody dependent cellular phagocytosis was also increased. The manuscript is well structured and accessible. The findings are really interesting and support the notion that nnAbs might play an important role in HIV treatment strategies. However, one major issue and a number of minor issues remain to be resolved before publication.
Major
The results suggest a prominent induction of antibody dependent cellular phagocytosis (ADCP) by the ID2-V1V2 construct. Unfortunately, the authors do not elaborate on the physiological significance of ADCP in HIV response. Since they observe pronounced differences between ID2 alone and ID2 + ID2-V1V2 in eliciting ADCP, the ADCP should be introduced more carefully and its relevance should be discussed in more detail.
Minor issues
Line 72: ADCC stands for “antibody dependent cellular cytotoxicity”. The sentence should be corrected.
Line 83: along the same line ADCP is the acronym for “antibody dependent cellular phagocytosis”.
Line 93: the resulting ID2-V1V2 fusion construct with linkers and additional cysteines is pretty complex. For transparency, the authors should provide the entire DNA sequence of the expressed fusion.
Line 223: typo, it should read “constitutes”.
Line 234: The “data not shown” must be either included or not mentioned.
Figure 1A: it is not clear why the asparagine residues (N) are shaded in grey. It should be explained in the figure legend.
Figure 1B: axis need to be labeled and the peaks of the standard need to be assigned.
Figure 2: the two panels should be labeled with a and b, to match the figure legend.
Line 337: it should read “reaching”.
Line 343: it is “sera”. Perhaps, the authors should run a spell check.
Figure 5: it is not obvious what was statistically compared, not is it clear how the number of stars relates to a specific p-value. The figure legend should include this information. More generally, the statistical procedures should be described in detail in the methods section.
Figure 6: again, the number of stars should be linked to specific p-values in the figure legend.
Line 380: the sentence starting in line 380 is confusing and should be rephrased.
Author Response
Response to reviewer 3
Sherburn et al. design a minimal HIV envelope construct to induce potent Fc dependent responses s. The authors previously presented data on a minimal structural unit comprising the inner domain antigen (ID2), a region that includes the CD4 induced epitope (i.e. upon CD4 binding the epitope becomes available, CD4i) from the constant region 1 & 2 (C1, C2) which is also referred to as region A. Some data suggests that in so far only partially successful vaccine trial RV144 with the V1V2 epitope, the antibody dependent cellular cytotoxicity (ADCC) by non-neutralizing antibodies (nnAbs) plays a role in reducing HIV infection. Likewise, the ID2 epitope has been shown by the authors to elicit non-neutralizing antibodies that induce strong Fc dependent responses. This suggests that both strategies result non-neutralizing antibodies that trigger ADCC dependent responses. Here, they reasoned that the two epitopes that were partially successful, ought to be combined in a structure that is based on their ID2. To this end, they engineered the V1V2 on top of their ID2 structural unit. The ID2-V1V2 showed poor responses to ID2, suggesting that the engineered V1V2 loop might shield the ID2 scaffold. However, they find that immunization with ID2-V1V2 together with ID2 induced a strong Fc-dependent response. The ADCC response was stronger than with their previous ID2 alone. Moreover, antibody dependent cellular phagocytosis was also increased. The manuscript is well structured and accessible. The findings are really interesting and support the notion that nnAbs might play an important role in HIV treatment strategies. However, one major issue and a number of minor issues remain to be resolved before publication.
We thank the reviewer for the favorable summary of the paper. Please find below a point by point response to any concerns raised. All changes to the text have been made as ‘track changes’
The results suggest a prominent induction of antibody dependent cellular phagocytosis (ADCP) by the ID2-V1V2 construct. Unfortunately, the authors do not elaborate on the physiological significance of ADCP in HIV response. Since they observe pronounced differences between ID2 alone and ID2 + ID2-V1V2 in eliciting ADCP, the ADCP should be introduced more carefully and its relevance should be discussed in more detail.
We thank the reviewer for this observation and apologize for the omission. This concern was also raised by another reviewer so a more thorough introduction, interpretation of results and discussion have been added to the manuscript. Changes have been made via ‘track changes’.
Line 72: ADCC stands for “antibody dependent cellular cytotoxicity”. The sentence should be corrected.
Line 83: along the same line ADCP is the acronym for “antibody dependent cellular phagocytosis”.
We thank the reviewer for noticing these errors, they have been corrected
Line 93: the resulting ID2-V1V2 fusion construct with linkers and additional cysteines is pretty complex. For transparency, the authors should provide the entire DNA sequence of the expressed fusion.
We agree with the reviewer that some sequence information would be beneficial for the readers. We have therefore made a supplementary figure containing the full length clade A/E 93TH057 gp120 amino acid sequence with regions used to make ID2-V1V2 highlighted. Residues Val65 and Ser115 which are mutated to cysteine in ID2-V1V2 to make a stabilizing disulfide bond are also highlighted. The figure also contains the amino acid sequence of clade A/E 93TH057 ID2-V1V2 colored with the same scheme as gp120 with the Gly-Gly linker that replaces the gp120 outer domain and the ID2-V1V2 stabilizing disulfide bond colored. For absolute transparency, the optimized DNA sequence of clade A/E 93TH057 ID2-V1V2 is included as panel C.
Figure 1 has also been altered to show ID2-V1V2 in the context of gp120 to clarify the origin of the immunogen.
Line 223: typo, it should read “constitutes”.
We thank the reviewer for noticing this, it has been corrected.
Line 234: The “data not shown” must be either included or not mentioned.
This has been removed
Figure 1A: it is not clear why the asparagine residues (N) are shaded in grey. It should be explained in the figure legend.
We thank the reviewer for noticing this. Figure 1 has been revised to make the origin of the immunogen more clear within the context of gp120 and no longer contains the shaded residues.
Figure 1B: axis need to be labeled and the peaks of the standard need to be assigned.
This has been corrected to label the axis, standard peaks and protein peaks.
Figure 2: the two panels should be labeled with a and b, to match the figure legend.
Figure 2 A has been altered for clarity to include two separate tables, they are now labeled A, B and C.
Line 337: it should read “reaching”.
Line 343: it is “sera”. Perhaps, the authors should run a spell check.
Typos have been corrected
Figure 5: it is not obvious what was statistically compared, not is it clear how the number of stars relates to a specific p-value. The figure legend should include this information. More generally, the statistical procedures should be described in detail in the methods section.
We thank the reviewer for noticing this omission. The figure legend has been altered to include the information that statistical analysis is between ID2 and ID2-V1V2 samples.
The number of stars are now explained in figure legends
Statistical procedures have been added to the methods section.
Figure 6: again, the number of stars should be linked to specific p-values in the figure legend.
This has been corrected, as above
Line 380: the sentence starting in line 380 is confusing and should be rephrased.
We thank the reviewer for pointing this out, the sentence has been re-worded to read ‘It is therefore also likely that any therapeutics using passive transfer of non- or weakly-neutralizing antibodies, which rely of Fc-effector activities for their function, will fail if only a single epitope is targeted.’

Round 2
Reviewer 1 Report
All the queries have been addressed.
Reviewer 2 Report
The manuscript has been improved.
In summary, it is of interested and conclusion are supported by the data.
Reviewer 3 Report
the authors addressed all my comments